# Childhood and Adolescent Central Nervous System Tumours in Spain: Incidence and Survival over 20 Years: A Historical Baseline for Current Assessment

**DOI:** 10.3390/cancers15245889

**Published:** 2023-12-18

**Authors:** Maria D. Chirlaque, Rafael Peris-Bonet, Antonia Sánchez, Ofelia Cruz, Rafael Marcos-Gragera, Gonzalo Gutiérrez-Ávila, José R. Quirós-García, Fernando Almela-Vich, Arantza López de Munain, Maria J. Sánchez, Paula Franch-Sureda, Eva Ardanaz, Jaume Galceran, Carmen Martos, Diego Salmerón, Gemma Gatta, Laura Botta, Adela Cañete

**Affiliations:** 1Consortium for Biomedical Research in Epidemiology and Public Health (CIBERESP), 28029 Madrid, Spain; mdolores.chirlaque@carm.es (M.D.C.); rmarcos@iconcologia.net (R.M.-G.); mariajose.sanchez.easp@juntadeandalucia.es (M.J.S.); me.ardanaz.aicua@navarra.es (E.A.); dsm@um.es (D.S.); 2Department of Epidemiology, Murcia Regional Health Authority, 30071 Murcia, Spain; antonia.sanchez20@carm.es; 3Department of Epidemiology, Regional Health Council, IMIB-Arrixaca, Murcia University, El Palmar, 30120 Murcia, Spain; 4Spanish Registry of Childhood Tumours (RETI-SEHOP), University of Valencia, Faculty of Medicine, 46010 Valencia, Spain; 5Neuro-Oncology Unit, Paediatric Cancer Centre, Sant Joan de Déu Hospital, Esplugues de Llobregat, 08950 Barcelona, Spain; ofelia.cruz@sjd.es; 6Epidemiology Unit and Girona Cancer Registry, Oncology Co-Ordination Plan, Catalonian Oncology Institute, 17004 Girona, Spain; 7Josep Trueta Girona Biomedical Research Institute (IDIBGI), Salt, 17190 Girona, Spain; 8Statistics, Econometrics and Health Research Group (GRECS), University of Girona, 17004 Girona, Spain; 9Josep Carreras Leukaemia Research Institute, 08916 Badalona, Spain; 10Castile-La Mancha Cancer Registry, Regional Health Authority, 45071 Toledo, Spain; ggutierrez@jccm.es; 11Asturias Cancer Registry, Public Health Directorate, 33006 Oviedo Asturias, Spain; joseramon.quirosgarcia@asturias.org; 12Valencian Regional Childhood Cancer Registry, Non-Communicable Disease Epidemiology and Surveillance Department, General Subdirectorate of Epidemiology and Health Surveillance, General Directorate of Public Health and Addictions, Regional Public Health Authority, Valencian Regional Authority, 46010 Valencia, Spain; almela_fer@gva.es; 13Basque Country Cancer Registry, Health Department, Basque Country Regional Authority, 01010 Vitoria-Gasteiz, Spain; arantza-lopez@euskadi.eus; 14Andalusian School of Public Health (EASP), 18011 Granada, Spain; 15Granada Bio-Health Research Institute, 18012 Granada, Spain; 16Mallorca Cancer Registry, General Directorate of Public Health and Participation, Balearic Isles Health Research Institute (IdISBa), 07010 Palma de Mallorca, Spain; pfranch@dgsanita.caib.es; 17Navarre Public Health Institute, 31003 Pamplona, Spain; 18IdiSNA, Navarre Health Research Institute, 31008 Pamplona, Spain; 19Tarragona Cancer Registry, Cancer Epidemiology and Prevention Service, Sant Joan de Reus University Teaching Hospital, 43204 Reus, Spain; jaume.galceran@salutsantjoan.cat; 20Pere Virgili Health Research Institute (IISPV), 43204 Reus, Spain; 21Faculty of Medicine and Health Sciences, Rovira i Virgili University (URV), 43003 Reus, Spain; 22Rare Diseases Research Unit, Foundation for the Promotion of Health and Biomedical Research in the Valencian Region (FISABIO), 46020 Valencia, Spain; carmen.martos@fisabio.es; 23Department of Health and Social Sciences, University of Murcia, 30100 Murcia, Spain; 24Evaluative Epidemiology Unit, Department of Epidemiology and Data Science, Fondazione IRCCS, Istituto Nazionale dei Tumori, 20133 Milan, Italy; gemma.gatta@istitutotumori.mi.it (G.G.); laura.botta@istitutotumori.mi.it (L.B.); 25Paediatric Oncology Department, La Fe Hospital, 46026 Valencia, Spain; 26Paediatrics, Obstetrics and Gynaecology Department, Faculty of Medicine, University of Valencia, 46010 Valencia, Spain

**Keywords:** childhood CNS tumours, adolescent CNS tumours, incidence, survival, time trends, paediatric oncology, population-based cancer registries, Spain

## Abstract

**Simple Summary:**

Central nervous system (CNS) tumours are highly common solid neoplasms in children and adolescents. Survival remains low in many countries, including Spain. While some studies have shown a rise in the incidence of these tumours in Europe, others have not. This study, the first in Spain, focused on two questions: (1) Is the incidence of CNS tumours increasing in Spanish children and adolescents? and (2) Has the survival of these patients improved? We analysed incidence in Spain across the period 1983–2007 and survival from 1991 to 2005, according to the International Childhood Cancer Classification. The incidence results revealed a stabilisation in children’s overall incidence trend since the early 1990s similar to that of Southern Europe. Overall survival was lower than that in Europe, without any improvement from 1991 to 2005. Our results provide a baseline for assessing current incidence and the achievements of paediatric oncology with regard to CNS tumours in children and adolescents.

**Abstract:**

Background: Central nervous system (CNS) neoplasms are highly frequent solid tumours in children and adolescents. While some studies have shown a rise in their incidence in Europe, others have not. Survival remains limited. We addressed two questions about these tumours in Spain: (1) Is incidence increasing? and (2) Has survival improved? Methods: This population-based study included 1635 children and 328 adolescents from 11 population-based cancer registries with International Classification of Childhood Cancer Group III tumours, incident in 1983–2007. Age-specific and age-standardised (world population) incidence rates (ASRws) were calculated. Incidence time trends were characterised using annual percent change (APC) obtained with Joinpoint. Cases from 1991 to 2005 (1171) were included in Kaplan–Meier survival analyses, and the results were evaluated with log-rank and log-rank for trend tests. Children’s survival was age-standardised using: (1) the age distribution of cases and the corresponding trends assessed with Joinpoint; and (2) European weights for comparison with Europe. Results: ASRw 1983–2007: children: 32.7 cases/10^6^; adolescents: 23.5 cases/10^6^. The overall incidence of all tumours increased across 1983–2007 in children and adolescents. Considering change points, the APCs were: (1) children: 1983–1993, 4.3%^ (1.1; 7.7); 1993–2007, −0.2% (−1.9; 1.6); (2) adolescents: 1983–2004: 2.9%^ (0.9; 4.9); 2004–2007: −7.7% (−40; 41.9). For malignant tumours, the trends were not significant. 5-year survival was 65% (1991–2005), with no significant trends (except for non-malignant tumours). Conclusions: CNS tumour incidence in Spain was found to be similar to that in Europe. Rises in incidence may be mostly attributable to changes in the registration of non-malignant tumours. The overall malignant CNS tumour trend was compatible with reports for Southern Europe. Survival was lower than in Europe, without improvement over time. We provide a baseline for assessing current paediatric oncology achievements and incidence in respect of childhood and adolescent CNS tumours.

## 1. Introduction

Tumours of the central nervous system (CNS) are the most common group of solid neoplasms in children. In Europe, age-adjusted incidence rates range from 38.9 (Northern Europe) to 30.0 (Eastern Europe) per million children aged 0–14 years (hereinafter, children) and from 36.2 (Northern Europe) to 20.8 (Eastern Europe) per million children aged 15–19 years (hereinafter, adolescents). The highest rates in the USA are for White non-Hispanics in both age ranges (38.2 and 24.8 cases per million children and adolescents, respectively). Incidence rates in other geographical areas, such as Africa and South Asia, are much lower [1]. While lower rates and relative variations in incidence may be a clue to ethnic variations [2,3,4], e.g., the distribution of risk factors in differing populations, they may also indicate a lower availability of diagnostic techniques [5,6,7]. CNS tumour survival has improved markedly since the 1980s for some types, such as embryonal tumours, but has changed little for others, such as astrocytic tumours [8], and remains limited in many countries, including Spain [9,10]. Brain vulnerability to therapies, especially at young ages, is a challenge for clinical management and means that many survivors of CNS tumours suffer serious sequelae that may imply lifetime medical surveillance, disabilities and an impaired quality of life [11].

CNS tumours are a heterogeneous set of neoplasms that arise at any age, from newborns to the elderly. Paediatric CNS tumours differ from adult CNS tumours in terms of origin, biology and anatomical site. This has challenged clinicians and epidemiologists, with different classifications over the years, based initially on pathology and currently on modern molecular techniques that reveal new entities [12,13]. For epidemiological purposes, the main diagnostic groups in children are ependymoma, astrocytoma, embryonal tumours and other gliomas [14]. There are large variations among these groups in terms of prognosis and response to therapies, even within the same diagnostic categories, e.g., ‘astrocytoma’ includes a variety of tumours, ranging from slow-growing pilocytic astrocytoma with good prognosis to malignant gliomas with very low survival [15,16]. Anatomical sites where the achievement of complete resection is difficult also affect prognosis in all CNS tumours, even those with “benign behaviour” [17]. The proportion of differing histologies and sites in a group may thus hinder the interpretation of survival results [18].

The increase in the incidence of childhood CNS tumours over time merits attention. In Europe, the incidence of CNS tumours in children increased over the period 1978–1997 by an average of 1.8% per year in malignant tumours (pilocytic astrocytoma included) and 1.7% per year in non-malignant tumours (pilocytic astrocytoma excluded) [19]. However, a more recent European study shows no rise in the incidence of malignant childhood CNS tumours (pilocytic astrocytoma excluded) from 1991 to 2010 in Northern, Eastern and Southern Europe (including Spain), though there was a rise in both Western Europe and Europe as a whole [20]. While differences between the two studies in terms of the tumours covered hinder comparability, the question of time trends in the incidence of CNS in children remains open.

This study on childhood and adolescent CNS tumours is an extension of the PITTI project [21,22,23,24], which was the first in the country to use population-based cancer registries (PBCRs) to study incidence, time trends and survival in childhood and adolescent cancer in Spain. It enjoyed the collaboration of all active PBCRs and the national childhood registry. The call for data closed in 2010.

The aim of this paper is to describe the incidence of and survival in childhood CNS tumours in Spain, including time trends, across the period 1983–2007 for incidence and 1991–2005 for survival. Our results are of interest for both epidemiology and clinics: (1) the incidence of CNS tumours in Spain and shifts therein over 25 years are shown; and (2) the survival results of this study will form the basis for assessing the current progress of clinical care for children with CNS tumours in Spain and the efforts made by clinicians and paediatric cancer centres to implement multidisciplinary protocols and European clinical trials.

## 2. Materials and Methods 

The cases to be included were defined as any child (<15 years of age) or adolescent (15–19 years of age) diagnosed with a CNS tumour (malignant or non-malignant) residing in the catchment areas of the 11 Spanish PBCRs collaborating in the PITTI study (Table 1), where the incidence occurred in the period 1983–2007. CNS tumours were defined as Group III of the International Classification of Childhood Cancer (ICCC-3) [25], which corresponds to the 3rd edition of the International Classification of Diseases for Oncology (ICD-O-3) [26]. Group III tumours are divided into six subgroups: IIIa. ependymomas and choroid plexus tumours; IIIb. astrocytomas; IIIc. intracranial and intraspinal embryonal tumours; IIId. other gliomas; IIIe. other specified intracranial and intraspinal neoplasms; and IIIf. unspecified intracranial and intraspinal neoplasms. Subgroups IIIa, IIIb, IIId, IIIe and IIIf include malignant and non-malignant tumours. Group III does not include germ cell tumours, which are therefore not the subject of this study. A detailed classification is shown in Appendix A, with sub-classifications for the extended subgroups of the ICCC-3 [25]. Ten cases with benign tumours located in the meninges, brain or spinal cord; the cranial nerves; or other parts of the CNS, which are excluded from Group III in the ICCC, were, however, present in the set of cases supplied by the participating PBCRs and were thus included in the analyses (see the added subgroup IIIe6 in Appendix A). Cases were extracted from the 11 participating PBCRs. A total of 1963 cases (malignant and non-malignant) were extracted from the 11 participating PBCRs, with all being used for the incidence study (1983–2007) and a subset of 1171 being used for the survival study (1991–2005). There were several reasons for limiting the survival study to the abovementioned subset. Firstly, prior to 1991, paediatric brain tumours in Spain were treated in different general and paediatric hospitals, either by neurosurgeons or paediatricians. The Spanish Society of Paediatric Oncology (*SEHOP*) and its Paediatric Brain Tumour Group developed guidelines and protocols from 1991 onwards aimed at improving clinical outcomes and also initiated intense collaboration with other European groups likewise aimed at improving results. Secondly, the use of magnetic resonance imaging (MRI) in children with brain tumours began to be implemented at that time, enabling diagnosis and evaluation of response to be performed more accurately than before. Thirdly, most cancer registries could not ensure a complete follow-up (for survival estimates) for cases incident in previous years.

The participating PBCRs share uniform data-collection procedures, with active case searching in multiple sources (including death certificates). Moreover, they all comply with the International Agency for Research on Cancer (IARC) [27], the International Association of Cancer Registries (IACR) [28] and the European Network of Cancer Registries (ENCR) [29] standards, and participate in international projects, such as ACCIS [30], EUROCARE [31], IICC-3 [32] and Cancer Incidence in Five Continents [33], which necessarily implies the appropriate quality ratings. All Spanish PBCRs form part of the REDECAN network [34]. The Spanish Registry of Childhood Tumours (*Registro Español de Tumores Infantiles* (*RETI-SEHOP*)) [35] was used to detect cases residing in the catchment areas of the regional registries but receiving care in other geographical areas. The 11 PBCRs covered 31% and 21% of Spanish children and adolescents, respectively (reference year: 2000 [36]).

The registration period of the participating PBCRs varied from 17 to 25 years, with at least half the registries covering the full 25-year period (Table 1). Non-malignant tumours were registered at 9 of the 11 contributing registries, while 2 PBCRs solely registered malignant tumours. Apart from the two cancer registries (CRs) that solely registered malignant tumours, the PBCRs varied widely when it came to registration of non-malignant tumours. This variability was present between registries and throughout the study period, with some CRs starting to register these tumours later than others. The proportion of non-malignant tumours across the study period rose from almost 14% in the first decade to around 30% in the last decade (Appendix A). Comparison of the largest registries which registered non-malignant tumours, as well as the increases across the period, suggests differences in registration criteria and the evolution of pathological and biological techniques for better categorisation. The proportion of non-malignant tumours—even in the last part of the period, when incidence was higher—was lower than could be expected from the figures reported for other European countries [37]. As regards adolescents, all CRs that registered non-malignant tumours recorded an average of <1 cases/year, except the Basque Country, with 2.6 cases/year. The Valencian Community CR did not provide data for adolescents.

All codes originally from the 1st and 2nd editions of the ICD-O were converted to those of the ICD-O-3 [26]. Data were verified using the DEP_Edits package to check the validity of codes and the internal consistency of the main variables (sex, site, age, date of birth and morphology) [38]. 

As mentioned above, for the survival analysis, we included registries with available data for the period 1991–2005 and follow-up until 31 December 2010. The Zaragoza CR, which could not meet the follow-up requirement, did not participate in the survival analysis. To obtain the most accurate information on vital status, all registries conducted follow-up procedures with common criteria, using the National Death Index (NDI) and other sources, such as regional mortality registries; the social security database; the patient referral files of the Autonomous Regions (*Comunidades Autónomas*), which make it possible to track patient referral and migration between regions; hospital clinical records; primary-care records; and municipal electoral rolls. Loss of follow-up among children resident in Spain at the time of tumour incidence is believed to be rare. On the other hand, systematic recording of treatment data was not available for most participating cancer registries.

Incidence rates were calculated as the annual number of cases per million child-/adolescent-years for all diagnostic subgroups of the ICCC-3 Group III and for the total of CNS tumours. Age-specific rates were obtained for the 0-, 1–4-, 5–9-, 10–14- and 15–19-years age groups. In order to obtain results that would be comparable over time and between countries, age-standardised rates were calculated for the 0–14- and 0–19-years age groups by the direct method, using the World Standard Population (ASRw) [39]. All rates were calculated for both sexes combined and for boys and girls separately. Populations at risk were obtained from the Spanish National Statistics Institute [36] and from the statistical institutes of the respective Autonomous Regions for Albacete, Girona, Granada, Mallorca, Tarragona and Zaragoza. 

Incidence time trends were analysed using the Joinpoint Regression Programme Version 4.1.1 [40,41]. To estimate the annual percent change (APC) for the period as a whole, the inflection points were limited to zero, and to assess if there were any change-points in the trend and the corresponding APCs of the relevant segments, two inflection points at most were allowed. When no cases were observed for a specific incidence year, 0.01 was imputed as the incidence rate value. Furthermore, the risk of developing CNS cancers (regardless of behaviour) was assumed to be similar across PBCRs.

Incidence and incidence time trends were analysed for Group III as a whole and all its subgroups. Due to the low number of cases, time-trend analyses were performed only for children (age-standardised incidence, 0–14 years) and adolescents. To avoid the effect of non-malignant tumours on time trends and to carry out comparisons with studies that solely addressed malignant tumours, time trends were also analysed for malignant tumours alone (behaviour code: /3).

Survival analyses were performed according to ICCC-3 Group III, the group as a whole and its subdivisions, and by age group. In addition, ependymomas (Subgroup IIIa1, ICD-O-3 morphology codes: 9383/1, 9391/3, 9392/3, 9393/3 and 9394/1) and medulloblastomas (IIIc1, ICD-O-3 morphology codes: 9470/3, 9471/3, 9472/3, 9474/3 and 9480/3) (Appendix A) were analysed separately due to their importance in clinical practice. This study covers a period when European collaborative trials on ependymoma and medulloblastoma were proposed and conducted by clinicians and Spanish paediatric oncology units were participating [42,43,44,45].

Observed 5-year survival was calculated for three cohorts by year of diagnosis (1991–1995, 1996–2000 and 2001–2005) using the Kaplan–Meier method [46]. Additionally, survival rates for 0–14 years of age were standardised by age, using the age distribution of the pool of cases in the survival analysis as standard. Where the 0-year age group had no cases, it was excluded from the calculation of standardised survival. No other age group displayed 0 cases. A log-rank test [47] was used to test the differences between survival curves by type of tumour and age group. The following were also used: a log-rank test for trend [48] to compare trends in observed survival estimates across the successive cohorts; and the APC calculated with the Joinpoint programme to characterise time trends in age-standardised survival (0–14 years) across the three cohorts. As in previous papers [19], results for observed or standardised survival for fewer than 10 cases are not shown because of the uncertainty of their estimates.

Furthermore, survival was estimated for some selected groups of tumours, such as all malignant tumours (behaviour ICD-O-3 code: /3), all non-malignant tumours (behaviour ICD-O-3 code: </3), pilocytic astrocytoma (ICD-O-3 code: 9421/1), astrocytoma NOS (ICD-O-3 code: 9400/3), malignant glioma (ICD-O-3 code: 9380/3, optical tract excluded) and lethal tumours (ICD-O-3 codes: 9401/3, 9440/3, 9441/3, 9442/3, 9451/3 and 9508/3), as defined in previous papers [37]. 

To compare Spanish 5-year survival results to those of Europe, we used the previous study by Gatta et al. [37] as a reference. This reports CNS 5-year survival results for Europe across the period 2000–2007 according to the ICCC-3 classification and thus enables Spanish results for the 2001–2005 cohort to be compared to European results. Since Gatta et al.’s survival estimates for total CNS, total malignant and total non-malignant tumours were age-standardised by the age distribution in the European pool, we standardised our 5-year survival results for these three groups using the same weights. Where appropriate, 4 PBCRs with <25% of non-malignant tumours were removed from the comparisons with Europe to improve comparability, in accordance with the criteria proposed by Gatta et al. [37]. In the Spanish results, tumours in the CNS diagnostic subgroups were also labelled with the WHO grading [49] applied by Gatta et al. for comparison with Europe (Appendix A). To compare the observed proportions of diagnostic subgroups in Spain to those reported in Europe, we performed chi-squared tests using R, with Yates correction for continuity being used to account for the low frequency of some morphologies. *P*-values were adjusted for multiple testing using the Benjamini and Hochberg method [50]. Other statistical analyses were performed with the SPSS version 28.0.1.1 and R version 4.2.2 software packages. The threshold for statistical significance was set at *p* < 0.05.

## 3. Results

### 3.1. Incidence

A total of 1963 cases were included in the incidence analysis for the period 1983–2007. The percentage of notified microscopic verification was 82.5% in children and 84.5% in adolescents (Table 1). The breakdown for the total of CNS tumours was as follows: malignant, 78.0% in children and 71.6% in adolescents; uncertain, 18.8% and 19.2%; and benign, 3.2% and 9.1%, respectively. A total of 1093 cases were diagnosed in boys and 870 in girls (Appendix A). The age group with the highest number of cases was 0–4, with cases decreasing with age (Appendix A).

The most frequently diagnosed subgroups were IIIb (42%) and IIIc (21%) in children and IIIb (44%) and IIIe (16%) in adolescents. By age group in children, the most frequently diagnosed subgroup was IIIb, accounting for at least 40% in all age groups (Table 2). By morphology, the most frequently diagnosed types were astrocytoma NOS (23%) in the 0–14-years age group (Figure 1A) and astrocytoma NOS (23%) in the 15–19-years age group (Figure 1B). The relative weights of tumours by morphology in the ICCC-3 subgroups for the total period 1983–2007 are shown in Appendix A. 

In children, the proportion of malignant tumours was greater than that of benign tumours throughout the five subperiods studied, ranging from 86% up until the early 1990s and falling to 68.5% between 1998 and 2002 (Figure 2, Appendix A). Over the entire period, a total of 84 lethal tumours were diagnosed: 58 cases at ages 0–14 years (3.5% of all CNS tumours) and 26 cases at ages 15–19 years (7.9%). The most frequently diagnosed group were malignant gliomas, formerly classified as glioblastomas, which accounted for 48.3% of lethal CNS tumours in children and 65.4% in adolescents.

### 3.2. Incidence Rates

The present analysis includes data from different regions of Spain to ensure consistency with respect to overall incidence at a national level. Although this issue was not the designated study objective, some variability was nonetheless in evidence between regions, as a north/south gradient or lower numbers of cases in different geographical areas. 

The crude incidence rates for all CNS tumours were 31.8 per million in children and 23.5 per million in adolescents (Table 3), with a boy:girl sex ratio of 1.26 in children and 1.22 in adolescents. The highest age-specific incidence rates for both sexes and for boys and girls separately were observed at ages 0–4 years (37.1, 41.3 and 32.7, respectively) and the lowest at ages 15–19 years (23.5, 25.2, and 21.7, respectively) (Table 3, Appendix A).

For both sexes combined, the highest incidence rate corresponded to Group IIIb in all age groups (Table 3 and Appendix A). Subgroups IIIa, IIIb, IIIc and IIId displayed the highest incidence at ages 0–4 years (5.3, 15, 8.6 and 4, respectively), and the subgroups IIIe and IIIf at ages 15–19 years, with figures of 3.7 and 3.2, respectively. 

The ASRw of CNS tumours in children for the entire period was 32.7, and in children and adolescents as a whole, it was 30.6 (Table 3), with a boy:girl sex ratio of 1.26 in both groups. The most frequently diagnosed group in children throughout the period was Group IIIb (13.5 in both, 13.7 in boys and 13.3 in girls), followed by IIIc (7.1 in both, 8.6 in boys and 5.5 in girls) (Table 3 and Appendix A).

### 3.3. Incidence Time Trends, 1983–2007

Figure 3 shows the age-specific incidence rates in children and adolescents for all CNS tumours across the period 1983–2007. Figure 4 shows the ASRws of CNS tumours in children (0–14 years) in Spain for the period 1983–2007 by diagnostic subgroup according to the ICCC-3; Figure 5A,B depict the Joinpoint analysis results for malignant and non-malignant tumours combined; and Figure 5C,D show the trends for malignant tumours only. 

Incidence rates of malignant and non-malignant tumours combined rose steadily across the entire study period for all CNS tumours in both children and adolescents (APC = 1.4%^ and APC = 2.4%^, respectively). Increases across the whole study period were also observed in children in Subgroups IIIa (APC = 2.3%^) and IIIe (APC = 6.0%^) and in adolescents in Subgroup IIIe (APC = 5.2%^). No significant trends were observed in other subgroups for the study period as a whole.

When change points were taken into account, significant increases were observed for total CNS tumours: in children, from 1983 to 1993 (APC = 4.3%^), with *k* = 1 (*k* = number of change points); and in adolescents, from 1983 to 2004 (APC = 2.9%^), with *k* = 1. Significant increases were likewise observed in children for the following subgroups: IIIa, from 1996 to 2007 (APC = 9.1%^), with *k* = 2; IIIb, from 1983 to 1997 (APC = 4.0%^), with *k* = 1; and IIIc, from 1987 to 2007 (APC = 2.2%^), with *k* = 1. Group IIIe showed positive trends in children from 1983 to 1998 (APC = 9.6%^), with *k* = 1, and in adolescents from 1991 to 2007 (APC = 6.9%^), with *k* = 1. 

No significant trends were observed in children: for all CNS tumours combined from 1993 to 2007; for Subgroup IIIa for the entire period 1983–2007, with *k* = 1, and from 1983 to 1996, with *k* = 2; for IIIb from 1997 to 2007, with *k* = 1, and for the entire period 1983–2007, with *k* = 2; for IIIc from 1983 to 1987, with *k* = 1, and for the entire study period, with *k* = 2; and for Subgroup IIIe from 1998 to 2007, with *k* = 1, and for the entire period 1983–2007, with *k* = 2. Subgroups IIId and IIIf showed no significant trends. 

In adolescents, no significant trends were present, with the exception of the above trend for all CNS tumours combined; the abovementioned rising trend in IIIe; and a significant downward trend across the period 1986–2007 for IIId, with *k* = 1 (APC = −2.6%^).

When the analysis was restricted to malignant tumours in children, no significant increases across the whole study period were observed for all CNS tumours combined or for any of the subgroups. Yet when the analysis was broken down by subgroup and change points were introduced, the position was as follows: IIIa showed a rising trend across the period 1998–2007 (APC = 11.5%^) with *k* = 2; IIIb, with *k* = 1, showed an increase (APC = 7.4%^) from 1983 to 1991, followed by a decrease (APC = −3.9%^) until the end of the study period, and similar results with *k* = 2; IIIc displayed a rising trend (APC = 2.2%^) from 1987 to 2007 with *k* = 1, and a fluctuating, albeit non-significant, trend with *k* = 2; IIIe and IIIf showed no significant trends (Figure 5C). Adolescents displayed no significant increases for the whole study period when cases were restricted to malignant tumours only. No trends were present for all CNS malignant tumours combined, and when analysed by subgroup, the number of cases in adolescents was very low (Figure 5C). 

### 3.4. Survival

The survival analysis included 1171 cases, 0–19 years of age, for the whole period 1991–2005. Of this total, 972 (83%) were children (0–14 years of age) and 199 (17%) were adolescents (15–19 years of age). The Zaragoza registry did not contribute to the survival analysis, while the Valencian Region Childhood Cancer Registry sent data solely for children. The 5-year follow-up was high, reaching 99% (Table 1)

Figure 6 and Table 4 show 5-year survival for CNS tumours in children and adolescents, overall and by diagnostic subgroup. For the period as a whole, the overall 5-year observed survival for all CNS tumours was 65% for children and adolescents alike. As expected, survival in children and adolescents was significantly different among the diagnostic subgroups: in children, 5-year observed survival ranged from 45% (other gliomas) to 83% (astrocytomas), and in adolescents, from 29% (unspecified) to 85% (other specified). At the same time, there were no statistically significant differences in survival between these two age groups for all CNS tumours, whether overall or by diagnostic subgroup, except in the case of astrocytomas. Marked differences for subgroups IIIa (ependymomas and choroid plexus tumours) and IIIf (unspecified) did not reach statistical significance.

The whole set of CNS tumours (Figure 7 and Table 5) showed no significant trends in the three cohorts considered for analysis (1991–1995, 1996–2000 and 2001–2005) either in children or adolescents, or in observed or standardised survival. Likewise, no differences were observed in the respective cohorts between these two groups of patients. 

Detailed results for children, broken down by cohort, are listed in Appendix A, which shows observed 5-year survival for CNS tumours by diagnostic subgroup and age group, as well as standardised 5-year survival for 0–14 years of age, log-rank results for differences by age and time trends of observed 5-year survival, and APCs for standardised survival. Since the number of cases was low and results for less than 10 cases are not shown, many cells are blank. The pattern by age group is unclear. No 5-year survival time trends by age group showed statistical significance. In addition, the Joinpoint for standardised survival did not yield statistically significant APCs for any of the diagnostic subgroups or for the complete set of CNS tumours (Table 6).

Insofar as adolescents were concerned (Appendix A), there were only 10 cases or more in each cohort for all CNS tumours together and astrocytomas. The trends were not statistically significant in either of these two diagnostic groups. 

Astrocytomas, the sole subgroup with enough cases in both children and adolescents to be split into cohorts, showed a statistically significant difference (*p* < 0.001) in observed 5-year survival between children and adolescents in the most recent cohort (2001–2005), with a higher survival rate among children (Table 7). No time trends were statistically significant.

In children, pilocytic astrocytoma did not show a statistically significant increase in survival across the period, and while the same applied to astrocytoma NOS, it nevertheless showed a continuous, though not statistically significant, increase in all three cohorts (Appendix A, respectively, and Table 6). 

With regard to special groups IIIa1 (ependymomas) and IIIc1 (medulloblastomas) in children aged 0–14 years, which had been analysed separately, the results by cohort and age group are shown in Figure 8. The number of cases of ependymomas was less than 10 in most cells, thus making it impossible to show statistical differences between age groups (table not shown). While observed 5-year survival did not increase across the period, 5-year standardised survival did increase across the three cohorts (44%, 45% and 54%), but the time trend was not statistically significant (APC = 11% (−23;62)). Medulloblastomas also had a low number of cases, with some blank cells in the three cohorts. Here, again, this made it impossible to show statistical differences between the age groups. 5-year survival (both observed and standardised) increased slightly in the three cohorts (Figure 8B and Table 8), albeit without statistical significance (Table 6).

Overall survival in the case of malignant and non-malignant tumours was clearly different in both children and adolescents (*p* < 0.0001). However, no differences between these age groups appeared for malignant or non-malignant tumours (Table 9). Time trends in the three cohorts were not significant, except for non-malignant tumours in children aged 5–9 years and for children overall (0–14 years) when survival was standardised (Appendix A and Table 6). 

There were few lethal tumours in our series, but survival was relatively high, reaching a 5-year observed survival of 53% (27;78) in the most recent cohort (Appendix A). Malignant gliomas, similarly few in number, failed to show any trend (Appendix A and Table 6).

Appendix A give a detailed breakdown of survival in children, by cohort and age group, for pilocytic astrocytoma, astrocytoma NOS, malignant and non-malignant tumours, lethal tumours, and malignant gliomas.

## 4. Discussion

### 4.1. Incidence

This is the first large-scale report on the incidence and survival for childhood and adolescent CNS tumours in Spain and provides a comprehensive descriptive epidemiological overview. The results are based on data drawn from 11 internationally accepted population-based cancer registries (see above). In most registries, the number of cases reported is very low, especially for adolescents. The low numbers of cases and their random variations could be the main reason for the variability in certain indicators shown in Table 1.

The ASRw for all CNS tumours in children in Spain (32.7 per million) was lower than those of other European countries, such as Nordic countries [51,52], and higher than those of some Southern European countries, such as Slovenia or Turkey [53]. Differences in incidence among European geographical regions decreased in the first decade of the 21st century, reaching similar figures [1]. Among adolescents, the incidence rate observed in Spain (23.5 per million) was lower than that of children and lower than those of Northern, Western and Southern Europe but higher than those of Eastern Europe [1]. While all these differences could be accounted for, not only by differences in registration (i.e., early recording of non-malignant tumours according to expert recommendations) [54,55], but also by health issues (i.e., different access to medical services or diagnostic techniques in different countries and regions), a real increase in incidence cannot be ruled out [56].

ICCC-3 subgroups IIIa, IIIb, IIIc and IIId were also higher in children [32], while incidence in subgroups IIIe and IIIf, in contrast, was higher in adolescents than in children. As in other countries, incidence of neuroepithelial CNS tumours decreases as age increases, while incidence of meningeal, pineal and unspecified tumours increases as age decreases [57].

In agreement with other studies [1,19,58], the incidence of CNS tumours as a whole proved slightly higher in boys than in girls, for children and adolescents alike, with the boy:girl incidence ratio for CNS tumours being globally observed to remain stable from 1983 to 2007 [58]. In general, childhood cancer is more frequent in boys than girls worldwide. It is clear that in low-income countries where this difference is greater, easier access to health systems for boys plays a pivotal role vis-à-vis girls, but this factor would presumably not have any influence in high-income countries [58,59]. Prenatal endogenous factors, such as tumour-suppressor genes, which elude X-inactivation [60], might be one, though not the sole, explanation. 

The most frequently diagnosed CNS tumours in children were astrocytomas (41.7% of cases) and CNS embryonal tumours (21.1%). This same pattern has been seen for the whole of Europe [19,32,37], Britain [61] and Nordic countries [52].

A decrease in the percentage incidence of malignant tumours and an increase in that of non-malignant tumours can be observed in both children and adolescents throughout the entire period 1983–2007, though in the case of children this trend was slightly reversed in the last subperiod, from 2003 to 2007 (Figure 2)—a development which may be related to the progressive implementation of the registration of non-malignant CNS tumours. In 1983, only 4 of the 11 participating PBCRs registered benign CNS tumours, while in the year 2003, there were 9 that did so.

A steady increase in the incidence rates of all CNS tumours—malignant and non-malignant combined—can be observed in both children (1.4% per year) and adolescents (2.4% per year) across the entire period 1983–2007. However, if *k* = 1, then a first stage of increased incidence becomes visible in both groups, followed by the stabilisation of the trend until 2007. An increase in total CNS tumour incidence throughout the period is consistent with the findings published in some studies, e.g., a 0.8% increase in Australia from 1983 to 2016 [62]. When non-malignant tumours were excluded, however, no rising trend was present, a finding in line with the absence of rising trends observed by Steliarova-Foucher et al. [20] from 1991 to 2010 in Southern Europe (including Spain) and in the North and East (despite which, there was a rising trend in Western Europe and Europe as a whole). In Spain, an increase in all CNS tumours occurred until the early 1990s in children and until just before the mid-2000s in adolescents. The progressive incorporation of new diagnostic techniques, essentially MRI, may have partly contributed to this increase in incidence up to the early 1990s. 

An increased incidence of ependymomas and choroid plexus tumours was also observed in children (2.3% per year). Although this trend disappears when *k* = 1, if *k* = 2, then the Joinpoint model shows a steep increase from the mid-1990s until the final period (APC = 9.1%^). Furthermore, when non-malignant tumours were excluded, a steep increase (APC = 11.5%^) remained in evidence from 1998 to 2007. These tumours are more frequent in children aged 0 to 4 years. Ependymomas, and malignant ependymomas (ependymoma NOS and anaplastic ependymoma) in particular, accounted for 84% of tumours in this group. 

There was a significant rise in the incidence trend for the subgroup of astrocytomas (malignant and non-malignant combined) in children from 1983 to 1997 (APC = 4%^), which levelled off thereafter. Similar results were observed in Australia from 1983 to 1993 (APC = 4.1% (0.5; 7.8)) [62]. When non-malignant tumours were removed from the analysis, astrocytomas showed a significant rise (APC = 7.4%^) from 1983 to 1991, followed by a significant fall and subsequent stabilisation until the end of the study period. Astrocytoma was the most common histological subgroup, accounting for 41.7% of all CNS tumours in this study, whereas the single most frequent histological type was astrocytoma NOS, accounting for 54.4% of astrocytomas and 22.7% of all histological types. Similar results have been reported in other countries, such as Britain, where this subgroup accounted for 41% of all registrations [61].

Pilocytic astrocytoma in children accounts for 25% of all astrocytomas, 10.3% of all histological types and 34% of all non-malignant tumours. In general, the incidence of pilocytic astrocytoma in our study was lower than that observed in other countries, such as Britain [61], where it accounts for half of all astrocytomas. Pilocytic astrocytoma was reclassified in ICD-O-3 as behaviour 1 (in previous editions, ICD-O-1 and ICD-O-2, it was considered behaviour 3). In our series, pilocytic astrocytomas recorded prior to ICD-O-3 as behaviour 3 were converted to behaviour 1, and as a result, two PBCRs which did not register non-malignant tumours lost this tumour for the entire study period.

Intracranial and intraspinal embryonal tumours, all malignant in our series, composed basically of medulloblastomas (accounting for 84.3% of cases in this subgroup in children and 80% in adolescents), did not show a significant increase across the whole period. However, these tumours displayed an upward trend in children from the late 1980s until the end of the period (APC = 2.2% (0.1; 4.5)). A more gradual rising trend in this subgroup was found in The Netherlands in patients under 18 years old (APC = 1.2% (0.1; 2.3)) [63], as well as in children in Nordic countries (APC = 0.97% (0.02; 1.94)) [52] and Australia (APC = 0.8% (0.2; 1.3)) [62]. Nevertheless, when *k* = 2, the three segments observed in our data showed no statistically significant changes and they roughly followed the irregular oscillations that can be seen for this subgroup in Figure 5A.

As regards Subgroup IIId (other gliomas, mostly malignant gliomas (accounting for 75% of cases in this group in children and 81% in adolescents)), no increase was observed in children, while a downward trend in adolescents could be observed from the late 1990s until the end of the period, which disappeared when non-malignant tumours were removed. This may be interpreted as an improvement in diagnosis across the study period. That said, however, there were very few cases of this subgroup in adolescents throughout the entire period, making the Joinpoint results difficult to evaluate.

Other specified intracranial and intraspinal neoplasms, mostly non-malignant tumours, experienced a greater increase in incidence among children and adolescents across the entire study period, due mainly to the rise in incidence until the late 1990s in children and from the early 1990s to the end of the period in adolescents. When non-malignant tumours were removed, the number of cases dropped sharply and there were no increases in incidence to be seen. Although improvements in diagnostic techniques and an increase in benign CNS tumour registration could explain these facts, a real rise in incidence cannot be ruled out. 

Unspecified neoplasms remained stable throughout the period in children and adolescents. This group reflects the complexity of achieving a histological diagnosis in some cases, where samples might be scarce, heterogeneity might be present and neuropathological expertise is the most important tool for achieving a complete and accurate final diagnosis.

In summary, our study observed an increase in the incidence of all CNS tumours, malignant and non-malignant combined, followed by a stabilisation in the early 1990s among children and just before the mid-2000s among adolescents. In children, the overall trend approximates the pattern of Subgroup IIIb (astrocytomas), the subgroup with the highest frequency. The stabilisation of the overall trend in children and the absence of any rising trend in children for all CNS tumours combined when non-malignant tumours were removed from the analysis is in line with the observations of Steliarova-Foucher et al. [20] for Southern Europe. Similarly, it is in line with observations in individual Spanish registries [64] and, despite methodological differences in the classification of tumours and some statistical aspects, with the overall results for children of a recent REDECAN study on CNS tumours trends (excluding non-malignant tumours) in Spain (1985–2015) relating to mobile phone use [65]. Most upward trends may be hypothetically explained by a combination of several factors: changes in registry practices; changes in classification and coding; advances in diagnostic techniques; and, notably, the progressive registration of non-malignant tumours. Even so, this in no way excludes the participation of underlying risk factors in the increases. The paucity of data may have also affected the results, especially in adolescents (who had an average of 13 cases per year). 

### 4.2. Survival

The overall survival for Group III-ICCC3 over this long time period (1991–2005) is in line with previous reports by different groups for different time periods and geographical areas [8,9,19,37,62,63,66,67,68], including some individual Spanish registries [64]. Examination of the differences between children and adolescents reveals a dissimilarity with some EUROCARE-5 [69] results for Europe in that while our results show no difference between the two groups, the EUROCARE study reports better survival in adolescents for malignant cases. Comparisons are difficult, however, owing to variations in incidence periods, cohorts analysed and geographical areas, as well as worldwide disparities in access to care and health systems. Moreover, small differences in survival may be due to differences in the methods of estimation used.

If the average European survival reported in Gatta et al.’s study [37], which covers the 2000–2007 incidence period, is taken as a reference and compared to the 2001–2005 cohort in our study, overall CNS tumour survival figures (both age-standardised with the same weights and with registries with <25% of non-malignant tumours removed from the Spanish data, as in Gatta et al.’s study), albeit close, are shown to be lower in Spain than in Europe (Table 10). Overall survival is influenced by the proportion of non-malignant tumours. Specifically, this proportion was 38.5% in European data versus 33.3% in the comparable Spanish data when Spanish registries with <25% of non-malignant tumours were removed (Appendix A). This may partially account for the overall difference. In this regard, it is a fact that while the differences between Spain and Europe in standardised survival for all malignant tumours combined and for all non-malignant tumours combined are not large, survival for non-malignant tumours is higher in Europe (Table 10). That said, however, a look at individual countries in recent reports shows that Spain continues to remain low in the rankings [9,68]. 

When analysed by Group III-ICCC3 subdivisions (not standardised, observed survival), the difference with European data showed a higher survival in Europe than in Spain for all diagnostic subgroups except astrocytomas (Table 10). 

With reference to Subgroup IIIa (ependymomas and choroid plexus tumours), there were very few tumours of the choroid plexus in the Spanish 2001–2005 cohort, only three cases (Appendix A), and thus their influence on overall Subgroup IIIa survival cannot be important. Consequently, the difference in Subgroup IIIa in favour of the European pool should largely be attributable to ependymomas. Whereas 5-year observed survival for ependymomas in Spain was 45% (27; 63) in the 2001–2005 cohort, survival for the same morphology codes in the European pool ranged from 61% (9392/3, anaplastic ependymoma) to 96% (9394/1, myxopapillary ependymoma) [37]. The proportion of anaplastic ependymoma (among ependymomas) is 38% in Spain, close to that in Europe (41%), and the proportion of NOS in Spain is 48%, similar to the figure of 49% registered in Europe. Hence, the fact that Spain has a lower ependymoma survival figure than Europe cannot be explained by Spain’s having a higher proportion of anaplastic ependymoma than Europe. We feel that non-adherence to European ependymoma trials and the lack of administrative resources to conduct paediatric clinical trials at that time may well explain these results. We trust that analysis of more recent cohorts will show that the ependymoma survival rate has improved as a result of participation in the European trial that included central pathology and surgical review. 

Subgroup IIIb (astrocytomas) showed better survival in Spain than in Europe. This is a subgroup that allows for non-malignant tumours. Since the proportion of such tumours in the European pool is larger than in Spain, the higher survival in Spain cannot be explained by the overall proportion of non-malignant tumours. Looking at the relative distribution of the different tumours within Subgroup IIIb (Appendix A), the rates for Grade I and II tumours are 70% in Europe but only 53% in Spain, while the rates for the much less frequent Grade III and IV tumours are 14% in Europe and 10% in Spain. Specifically, pilocytic astrocytoma accounts for 53% of astrocytomas in Europe but only 31% in Spain (a statistically significant difference), and astrocytoma NOS accounts for 37% in Spain versus 16% in Europe (again significant). It is conceivable that astrocytoma NOS in Spanish data might conceal a proportion of pilocytic and other low-grade tumours that offset the differences indicated above, which, taken together with the difference in Grade III and IV tumours, could, at least in part, explain the higher survival in Spain. This explanation cannot, however, be proved with the data available in our study and would therefore call for further research beyond the scope of the present study. 

From a clinical standpoint, it is important to note that we found a statistical difference in the outcome of astrocytomas by age (0–14 versus 15–19 years) in favour of childhood cases in the most recent cohort (Table 7). The reasons for this, in spite of the low number of cases, might lie in a different disease biology according to the age spectrum [70] and/or different clinical management according to whether the patient is treated in a paediatric (more centralised care) or adult medical oncology setting (more dispersion), as occurs in Spain [71]. As rare diseases, centralisation of diagnosis and treatment is related to increased survival. It is also important to point out that, from 1991 to 2005, there were different reviews of the WHO classification for CNS tumours [72], which might have resulted in different diagnoses according to the new pathology classifications and, therefore, in a different coding of cases and impact on survival, as well as the LGG SIOPE trial dating from 2004 (pilocytic astrocytoma I° 9421/1, subependymal giant cell astrocytoma I° 9384/1, dysembryoplastic neuroepithelial tumour I° 9413/0, desmoplastic infantile ganglioglioma I° 9412/1, ganglioglioma I° and II° 9505/1, pleomorphic xanthoastrocytoma II° 9424/3, oligodendroglioma II° 9450/3, oligoastrocytoma II° 9382/3, astrocytoma II° 9400/3, fibrillary astrocytoma II° 9420/3, protoplasmatic astrocytoma II° 9410/3, and gemistocytic astrocytoma II° 9411/3, 0–16 years complete) [73].

Survival in astrocytoma NOS showed a slight increase in children across the period 1991–2005 (Appendix A), with its incidence (ASRw) decreasing from 10.0 per million children in the first cohort to 7.4 and 5.1 per million children in the second and third cohorts, respectively. Despite the fact that the increased survival for astrocytoma NOS was not statically significant, these data might reflect an improvement in diagnosis and a better classification of tumours with poor prognosis.

In medulloblastoma, the most frequent malignant childhood CNS cancer in Spain, slow, though not statistically significant, improvements were observed in 5-year age-standardised survival for the three successive cohorts (Table 8). Indeed, in the clinical field, it is very young children who need greater efforts to improve survival: disease aggressiveness and hurdles in treatment avoidance or deferral of radiation therapy due to its severe late CNS toxicity at this age, which hampers the quality of life of long-term survivors, make this a challenge for us.

Subgroup IIIe (other specified intracranial and intraspinal neoplasms) showed lower survival in Spain than the European average. This is a subgroup that includes many non-malignant tumours. Hence, the difference could also be explained in part by the bigger proportion of non-malignant tumours in Europe, as mentioned above [37].

A relevant point from a clinical point of view is the lack of improvement in survival in children and adolescents over the course of the three chronological cohorts (Table 5). This is in contrast to the position in Europe [67] and in other paediatric cancers in Spain, such as childhood leukaemias [23]. A multidisciplinary approach in centralised units, coupled with participation in clinical trials for paediatric cancers, is essential to improve survival in Spain. While the establishment and organisation of paediatric oncology units started as far back as 1990, the process is not yet complete because Spain has a decentralised health system with 17 different regions and more than 40 paediatric oncology units, many with a low number of cases, restricting the development of multidisciplinary expertise.

This study on childhood and adolescent CNS tumours is the first of its kind to be conducted in Spain, integrating all PBCRs and the national childhood registry. It gives a picture of the problem until the mid-2000s. However, several limitations must be considered:-One of the main limitations was the paucity of data, mainly for adolescents, which often made it difficult to obtain meaningful results when dividing the cases by type of tumour and age and/or sex;-A second limitation was the limited coverage of the 11 PBCRs that were the sources of the cases: the *RETI-SEHOP* was used solely to detect cases residing in the catchment areas of the regional registries but cared for in other geographical areas. These population-based registries covered 31% and 21% of the Spanish population aged 0–14 and 15–19 years, respectively. This fact restricts the representativeness vis-à-vis Spain as a whole;-A third limitation was the uneven registration of non-malignant tumours across the study period and between PBCRs;-Another serious limitation was the underlying complexity of the classification of CNS tumours and the changes made to it across the period [72]; the frequent difficulties of pathological diagnosis [74], which in Spain may be compounded by the dispersion of hospital units attending children and adolescents with CNS tumours; and the variability of registry practices with respect to CNS tumours—a phenomenon that has been observed worldwide [75]. In our study, a certain variability can be seen in the percentages of pilocytic astrocytoma, astrocytoma NOS, gliomas, lethal tumours and unspecified tumours among the participating population-based registries (Table 1), which, despite the small numbers, may be an expression of the abovementioned variability in registry practices. Classification-related problems may have led to exchanges between the different subgroups across the study period and may have created difficulties in interpreting the results for specific groups of tumours in terms of survival and incidence time trends. The progressive incorporation of non-malignant tumours into those eligible for registration and the different registration criteria applied to these tumours should also be borne in mind as a source of problems for the interpretation of results;-When it comes to survival, the quality of follow-up is decisive. An indicator of follow-up completeness is 5-year survival in lethal tumours. We used the same criteria (see the list of tumours in the Materials and Methods section) as Gatta et al., cited above [37]. In that study, overall 5-year survival for these tumours in Europe was 19%, whereas in Spain, 5-year observed survival in CNS lethal tumours was 33% in the first cohort, rising to 53% in the last (Appendix A), with very wide 95% CIs. These high survival proportion could, however, indicate some deficit in access to the data required to verify vital status and, as a result, some overestimation of 5-year survival.

## 5. Conclusions

We sought to analyse our findings by combining epidemiological and clinical reasoning in order to fully understand our results and any differences vis-à-vis other countries. The main findings are that CNS tumour incidence in Spain is similar to that in Europe and that rises in incidence may be mostly attributable to changes in the registration of non-malignant tumours. Without excluding the participation of underlying risk factors in the increases, the absence of a rise in the overall incidence time trend for malignant childhood tumours is compatible with reports on Southern Europe. Survival in Spain is lower than the European average, with no improvement over time across the period 1991–2005. Paediatric brain tumours are rare diseases, with all the related drawbacks, ranging from difficult diagnosis and changes in classification over time with scientific advances, to the need for a high level of therapeutic expertise. Indeed, they are the main example of a situation where centralisation of diagnosis and treatment is clearly related to increased survival. The lack of improvement over time in Spain may thus reflect this lack of centralisation over the years from 1991 to 2005. The results of this study provide a “historical baseline”—the best available for Spain—which, together with the current results of *RETI-SEHOP*’s specific analyses and the updated contributions of regional PBCRs, will make it possible to evaluate the current status of childhood CNS tumour incidence and the achievements of paediatric oncology in Spain. Such an endeavour would greatly benefit from a harmonisation of the registration practices for CNS tumours, including non-malignant tumours.

## Figures and Tables

**Figure 1 cancers-15-05889-f001:**
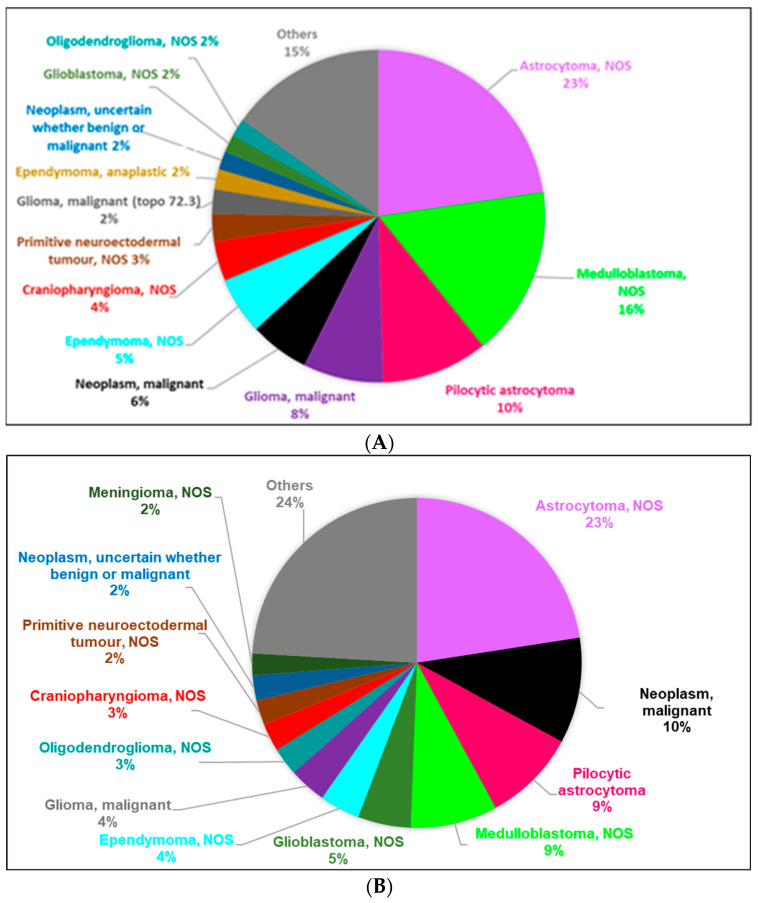
Distribution of number of new cases of CNS tumours (malignant and non-malignant combined), according to ICCC-3 Group III morphology (defined in the ICD-O-3), by age group for the period 1983–2007. (**A**) Children: patients aged 0–14 years. (**B**) Adolescents: patients aged 15–19 years. CNS: central nervous system (Group III of the ICCC-3); ICCC-3: International Classification of Childhood Cancer, third ed. [25]; ICD-O-3: International Classification of Diseases for Oncology, third ed. [26].

**Figure 2 cancers-15-05889-f002:**
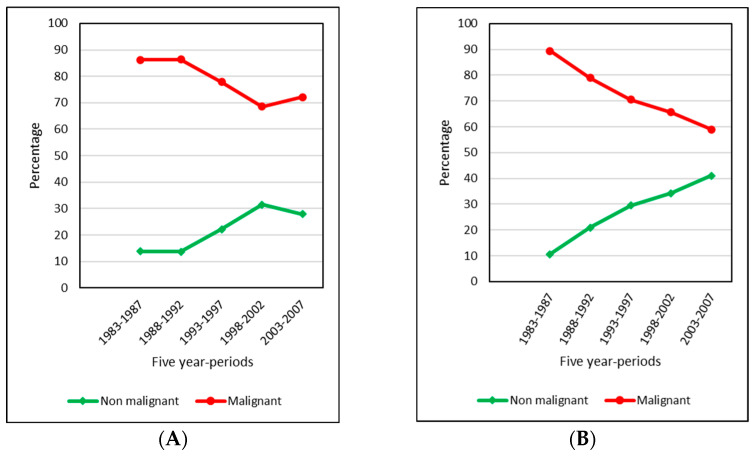
Percentages of malignant and non-malignant CNS tumour cases, according to Group III of the ICCC-3, (**A**) in children (0–14 years) and (**B**) in adolescents (15–19 years) for five five-year periods. Behaviour defined as in the ICD-O-3. ICCC-3: International Classification of Childhood Cancer, third ed. [25]; ICD-O-3: International Classification of Diseases for Oncology, third ed. [26].

**Figure 3 cancers-15-05889-f003:**
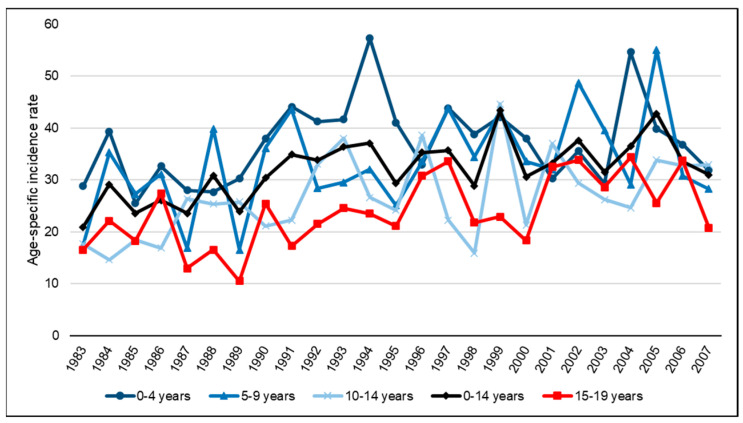
Incidence rates per million children or adolescents by age group for all CNS tumours (malignant and non-malignant combined), according to the ICCC-3, in Spain: 1983–2007.

**Figure 4 cancers-15-05889-f004:**
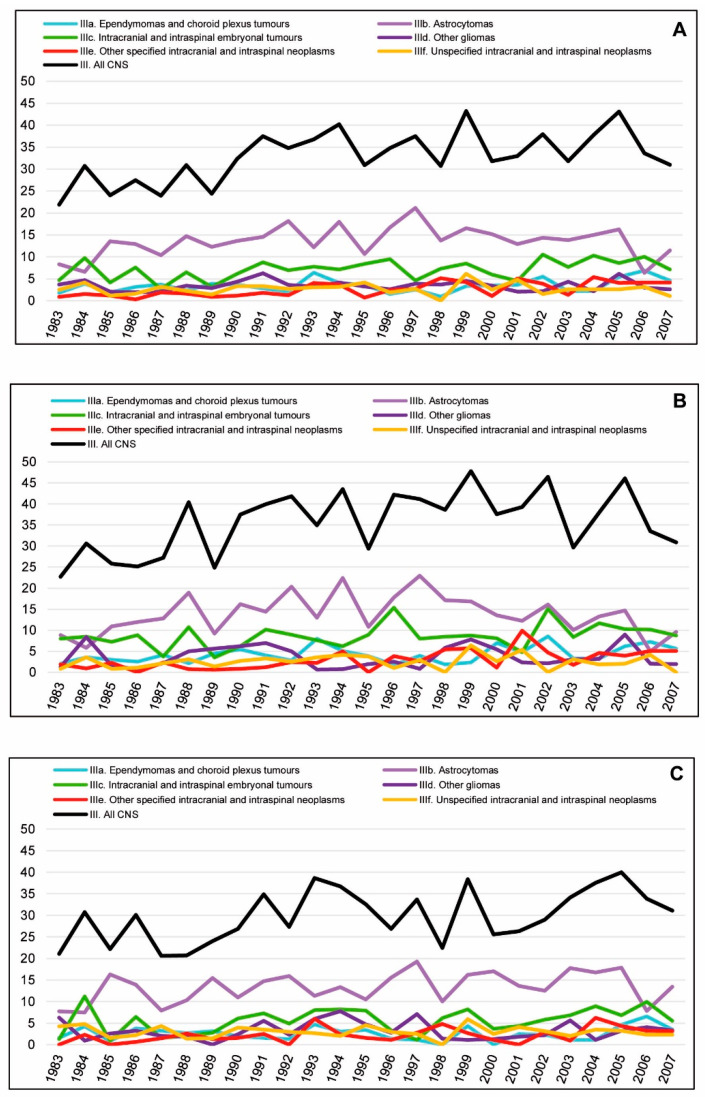
Trends in the incidence of CNS tumours (malignant and non-malignant combined) in children (0–14 years) in Spain, 1983–2007, by diagnostic subgroup according to the ICCC-3 and sex. (**A**) Both sexes. (**B**) Boys. (**C**) Girls. Age-standardised rates, World Standard Population (ASRw) per million children year; CNS: central nervous system (Group III of the ICCC-3); ICCC-3: International Classification of Childhood Cancer, third ed. [25].

**Figure 5 cancers-15-05889-f005:**
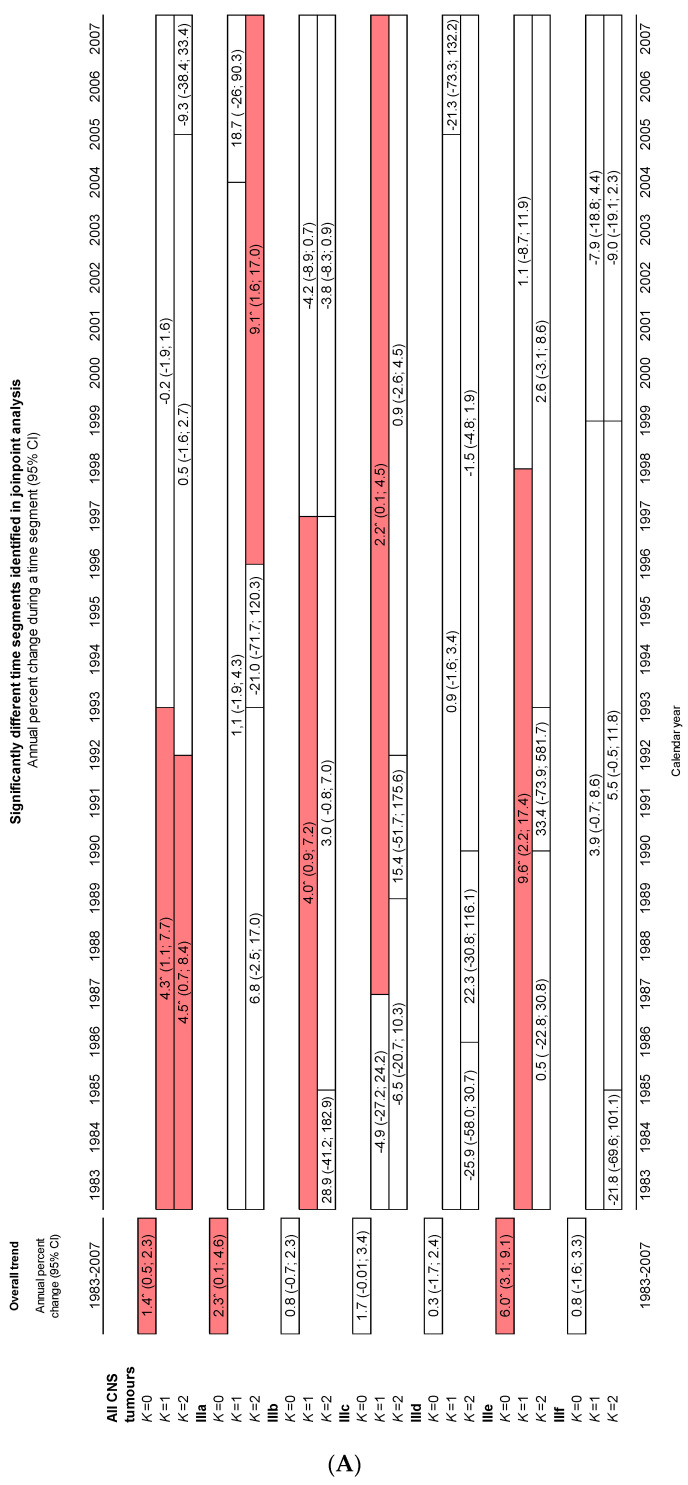
Observed incidence time trends of CNS tumours for the study period 1983–2007 by diagnostic subgroup, according to the ICCC-3. Joinpoint analysis. (**A**) Malignant and non-malignant tumours combined, children. (**B**) Malignant and non-malignant tumours combined, adolescents. (**C**) Only malignant tumours, children. (**D**) Only malignant tumours, adolescents. Red indicates an upward trend, blue a downward trend and white no significant trend. Absence of bars indicates that no joinpoint was identified for a given category. *k*: number of change points. ^ Statistically significant trend.

**Figure 6 cancers-15-05889-f006:**
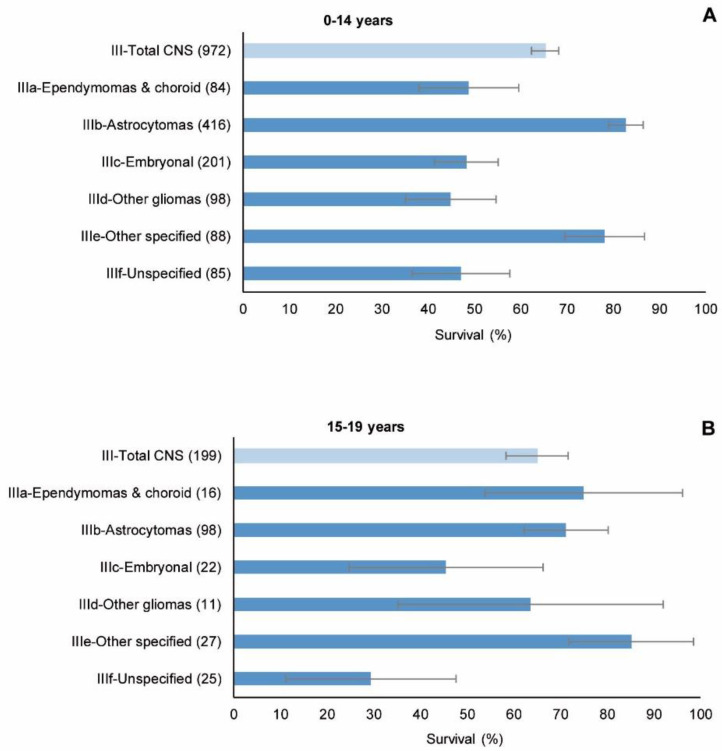
CNS tumours (malignant and non-malignant combined), period 1991–2005. 5-year observed survival by diagnostic subgroup of Group III of the ICCC-3. (**A**) Children. (**B**) Adolescents. Number of cases shown in brackets. ICCC-3: International Classification of Childhood Cancer, third ed. [25].

**Figure 7 cancers-15-05889-f007:**
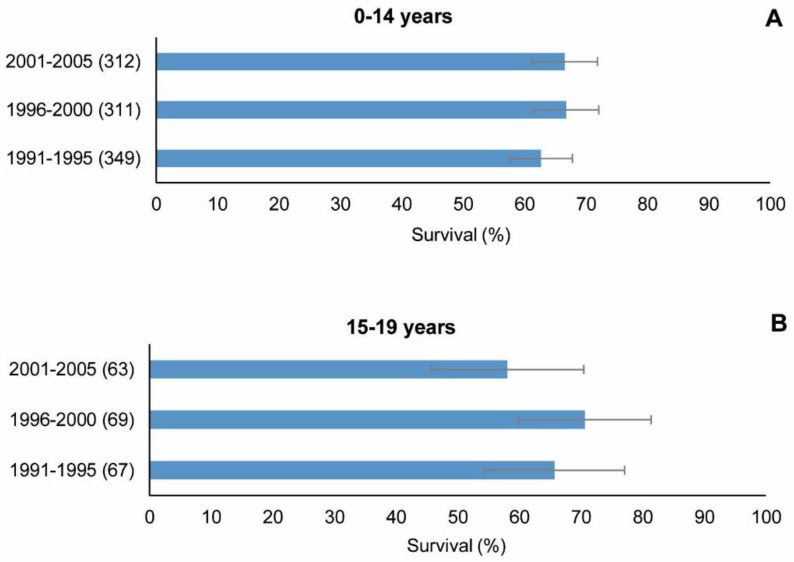
CNS tumours (malignant and non-malignant combined), period 1991–2005. 5-year observed survival by cohort of diagnosis and age group. (**A**) Children. (**B**) Adolescents. CNS: central nervous system (Group III of the ICCC-3). Number of cases shown in brackets. ICCC-3: International Classification of Childhood Cancer, third ed. [25].

**Figure 8 cancers-15-05889-f008:**
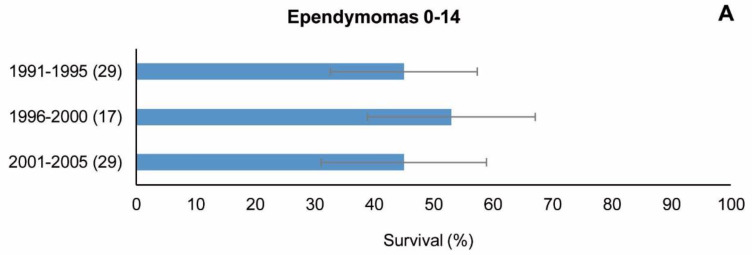
(**A**) Ependymomas (ICD-O-3 morphology codes: 9383/1, 9391/3, 9392/3, 9393/3 and 9394/1) and (**B**) medulloblastomas (codes: 9470/3, 9471/3, 9472/3, 9474/3 and 9480/3) in children, period 1991–2005. 5-year observed survival by cohort of diagnosis. Number of cases shown in brackets. ICD-O-3: International Classification of Diseases for Oncology, third ed. [26].

**Table 1 cancers-15-05889-t001:** Spanish population-based cancer registries (PBCRs) contributing to the study on CNS tumours, with details on their participation in the different analyses; periods of participation; person-years; numbers of cases; percentages of non-malignant, quality indicators; percentages of selected types of tumours in the subset for survival; and percentages of 5-year follow-up.

Registry	Incidence Analysis						Survival Analysis (Period of Incidence: 1991–2005)							
	Period	Period Length (Years)	Person-Years	n	Non-Malignant (%)	MV (%)	n	Non-Malignant (%)	Pilocytic (%)	Glioma (%)	Lethal (%)	IIIf (%)	Astrocytoma NOS (%)	Follow-Up ≥ 5 Years (%)
0–14 years														
Albacete	1991–2007	17	1,067,452	19	0.0	84.2	15	0.0	0.0	13.3	13.3	0.0	33.3	93.3
Asturias	1983–2007	25	3,859,144	92	18.5	83.7	49	28.6	22.4	16.3	2.0	2.0	16.3	100.0
Balearic Is *	1988–2007	20	1,955,526	62	24.2	79.0	49	28.6	18.4	6.1	4.1	16.3	14.3	98.0
Basque Country	1986–2007	22	6,715,925	267	30.3	80.5	172	34.3	12.8	2.3	4.1	19.2	14.5	100.0
Girona	1983–2007	25	2,336,906	89	27.0	87.6	57	29.8	15.8	14.0	5.3	5.3	15.8	93.0
Granada	1985–2007	23	3,719,973	87	25.3	82.8	54	27.8	24.1	1.9	3.7	14.8	11.1	100.0
Murcia	1983–2007	25	5,792,723	139	0.0	86.3	97	0.0	0.0	10.3	3.1	4.1	29.9	100.0
Navarre	1983–2007	25	2,207,823	81	21.0	82.7	47	21.3	17.0	10.6	14.9	6.4	12.8	100.0
Tarragona	1983–2007	25	2,531,824	64	15.6	82.8	29	17.2	17.2	3.4	0.0	20.7	20.7	100.0
Valencian Community	1983–2007	25	18,101,912	624	23.1	79.3	403	24.8	8.9	9.2	1.7	4.7	27.5	100.0
Zaragoza	1983–2006	24	3,163,796	111	26.1	96.4	-	-	-	-	-	-	-	-
All registries			51,453,004	1635	22.0	82.5	972	24.1	11.6	8.1	3.5	8.7	21.8	99.4
15–19 years														
Albacete	1991–2007	17	432,893	7	28.6	57.1	7	28.6	0.0	0.0	14.3	14.3	28.6	100.0
Asturias	1983–2007	25	1,774,905	30	43.3	90.0	16	37.5	12.5	0.0	0.0	6.3	37.5	100.0
Balearic Is *	1988–2007	20	798,482	18	22.2	94.4	15	20.0	20.0	0.0	40.0	6.7	20.0	100.0
Basque Country	1986–2007	22	3,124,106	93	43.0	81.7	66	43.9	19.7	3.0	6.1	12.1	12.1	100.0
Girona	1983–2007	25	905,361	20	25.0	75.0	17	23.5	17.6	5.9	5.9	23.5	17.6	88.2
Granada	1985–2007	23	1,500,706	25	36.0	88.0	15	33.3	20.0	0.0	20.0	13.3	20.0	100.0
Murcia	1983–2007	25	2,257,232	49	0.0	93.9	28	0.0	0.0	14.3	3.6	10.7	35.7	100.0
Navarre	1983–2007	25	917,627	26	38.5	80.8	19	52.6	5.3	0.0	0.0	10.5	15.8	100.0
Tarragona	1983–2007	25	1,007,151	25	0.0	76.0	16	0.0	0.0	0.0	6.3	18.8	43.8	100.0
Valencian Community	-	-	-	-	-	-	-	-	-	-	-	-	-	-
Zaragoza	1983–2003	21	1,250,000	35	28.6	85.7	-	-	-	-	-	-	-	-
All registries			13,968,463	328	28.4	84.5	199	29.6	12.6	3.5	8.5	12.6	22.6	99.0

CNS: central nervous system (Group III of the ICCC-3); DCO: death certificate only. There are 6 DCO cases: 3 in the 0–14-years age group (1 in Basque Country, 1 in Granada and 1 in Zaragoza) and 3 in the 15–19-years age group (1 in Asturias, 1 in Granada and 1 in Zaragoza). Non-malignant: 5th digit of the ICD-O-3 morphology code < 3; Pilocytic: pilocytic astrocytoma; NOS: not otherwise specified; Glioma: glioma NOS (ICD-O-3 morphology code: 9380/3) in all sites except optic nerve (ICD-O-3 code C72.3); Lethal: lethal CNS tumours (ICD-O-3 morphology codes: 9508/3 (atypical teratoid/rhabdoid tumour), 9401/3 (anaplastic astrocytoma), 9451/3 (anaplastic oligodendroglioma) and 9440/3-9442/3 (glioblastoma)); IIIf: unspecified NOS tumours of the CNS. Follow-up ≥ 5 years: percentage of patients with documented death before the 5th anniversary plus patients with documented vital status alive at the 5th anniversary of diagnosis or later with respect to all patients incident in the survival study period. The Zaragoza registry was not included in the survival analysis. The Valencian Region registry was not included in the analyses for the 15–19-years age group. ICCC-3: International Classification of Childhood Cancer, 3rd ed. [25]; ICD-O-3: International Classification of Diseases for Oncology, 3rd ed. [26]. * Majorca.

**Table 2 cancers-15-05889-t002:** Number of cases and percentages of CNS tumours (malignant and non-malignant combined) by age group, according to the ICCC-3, in Spanish children and adolescents across the period 1983–2007.

ICCC-3 CNS Group	Age 0–4		Age 5–9		Age 10–14		Age 15–19		Age 0–14		Age 0–19	
	N	%	N	%	N	%	N	%	N	%	N	%
IIIa. Ependymomas and choroid plexus tumours	81	14.2	44	7.9	38	7.5	20	6.1	163	10.0	183	9.3
IIIb. Astrocytomas	231	40.5	221	39.8	230	45.2	145	44.2	682	41.7	827	42.1
IIIc. Intracranial and intraspinal embryonal tumours	133	23.3	135	24.3	77	15.1	40	12.2	345	21.1	385	19.6
IIId. Other gliomas	61	10.7	59	10.6	50	9.8	26	7.9	170	10.4	196	10.0
IIIe. Other specified intracranial and intraspinal neoplasms	25	4.4	46	8.3	65	12.8	52	15.9	136	8.3	188	9.6
IIIf. Unspecified intracranial and intraspinal neoplasms	40	7.0	50	9.0	49	9.6	45	13.7	139	8.5	184	9.4
III. All CNS	571	100.0	555	100.0	509	100.0	328	100.0	1635	100.0	1963	100.0

Children: patients aged 0–14 years; Adolescents: patients aged 15–19 years; CNS: central nervous system (Group III of the ICCC-3); ICCC-3: International Classification of Childhood Cancer, third ed. [25].

**Table 3 cancers-15-05889-t003:** Incidence of CNS tumours (malignant and non-malignant combined) per million children/adolescents, according to the ICCC-3, by age group in Spain across the period 1983–2007. Age-specific rates and age-standardised rates (ASRws), World Standard Population.

ICCC-3 CNS Group	Age-Specific Rates					ASRws	
	0–4	5–9	10–14	0–14	15–19	0–14	15–19
IIIa. Ependymomas and choroid plexus tumours	5.3	2.6	2.0	3.2	1.4	3.5	3.0
IIIb. Astrocytomas	15.0	13.1	12.0	13.3	10.4	13.5	12.8
IIIc. Intracranial and intraspinal embryonal tumours	8.6	8.0	4.0	6.7	2.9	7.1	6.1
IIId. Other gliomas	4.0	3.5	2.6	3.3	1.9	3.4	3.1
IIIe. Other specified intracranial and intraspinal neoplasms	1.6	2.7	3.4	2.6	3.7	2.5	2.8
IIIf. Unspecified intracranial and intraspinal neoplasms	2.6	3.0	2.6	2.7	3.2	2.7	2.8
III. All CNS	37.1	32.9	26.5	31.8	23.5	32.7	30.6

ICCC-3: International Classification of Childhood Cancer, third ed. [25].

**Table 4 cancers-15-05889-t004:** CNS tumours (malignant and non-malignant combined) in children and adolescents, period 1991–2005, 5-year observed survival and follow-up by diagnostic subgroups of the ICCC-3 and age group. Log-rank results comparing the equality of survival distributions by diagnostic subgroup and age group.

Diagnostic Subgroup	Age Group				Log-Rank (*p*)
	Children		Adolescents		
	n	5-y Observed Survival (%) (95% CI)	n	5-y Observed Survival (%) (95% CI)	
IIIa. Ependymomas and choroid plexus tumours	84	49 (38; 60)	16	75 (54; 96)	0.071
IIIb. Astrocytomas	416	83 (79; 87)	98	71 (62; 80)	0.010
IIIc. Embryonal	201	48 (41; 55)	22	46 (25; 66)	0.926
IIId. Other gliomas	98	45 (35; 55)	11	64 (35; 92)	0.278
IIIe. Other specified	88	78 (70; 87)	27	85 (72; 99)	0.436
IIIf. Unspecified	85	47 (37; 58)	25	29 (11; 48)	0.063
III. Total CNS	972	65 (62; 68)	199	65 (58; 72)	0.952
Log-rank (*p*)		<0.001		<0.001	

CNS: central nervous system (Group III of the ICCC-3); ICCC-3: International Classification of Childhood Cancer, third ed. [25]; n: number of cases; 5 y: 5 years; (95% CI): confidence interval.

**Table 5 cancers-15-05889-t005:** CNS tumours (malignant and non-malignant combined) in children and adolescents, period 1991–2005. 5-year observed survival by cohort of diagnosis and age group, and standardised survival for children.

Cohort of Diagnosis	Age Group					Log-Rank (*p*)
	Children			Adolescents		
	n	5-y Survival (%) (95% CI)		n	5-y Observed Survival (%) (95% CI)	
		Observed	Standardised			
1991–1995	349	63 (58; 68)	62 (57; 67)	67	66 (54; 77)	0.601
1996–2000	311	67 (62; 72)	67 (62; 72)	69	71 (60; 81)	0.589
2001–2005	312	67 (61; 72)	67 (62; 72)	63	58 (46; 70)	0.202
Total	972			199		
Log-rank trend (*p*)		0.260			0.380	
Standardised survival trend APC (95% CI)			4 (−19; 34)			

CNS: central nervous system (Group III of the ICCC-3); ICCC-3: International Classification of Childhood Cancer, third ed. [25]; n: number of cases; 5 y: 5-year; (95% CI): confidence interval; APC: annual percent change.

**Table 6 cancers-15-05889-t006:** CNS tumours (malignant and non-malignant combined) in children by diagnostic subgroups of the ICCC-3 and selected tumours. Annual percent change (APC) for standardised 5-year survival trends across the three cohorts: 1991–1995, 1996–2000 and 2001–2005.

Diagnostic Subgroup and Selected Tumours	n	Standardised Survival Trend APC (95% CI)
III. Total CNS	972	4 (−19; 34)
IIIa. Ependymomas and choroid plexus tumours	84	8 (−26; 58)
IIIb. Astrocytomas	416	6 (−15; 31)
IIIc. Embryonal	201	4 (−49; 112)
IIId. Other gliomas	98	−14 (−87; 455)
IIIe. Other specified	201	4 (−60; 170)
IIIf. Unspecified	85	4 (−37; 72)
Ependymomas ^a^	75	11 (−23; 62)
Medulloblastomas ^b^	160	11 (−32; 80)
Pilocytic astrocytoma ^c^	113	−1 (−19; 21)
Astrocytoma NOS ^d^	212	6 (−8; 22)
Malignant ^e^	738	1 (−8; 11)
Non-malignant ^f^	234	4 ^ (2; 6)
Gliomas ^h^	79	−14 (−89; 581)

CNS: central nervous system (Group III of the ICCC-3); ICCC-3: International Classification of Childhood Cancer, third ed. [25]; n: number of cases; (95% CI): confidence interval. ^a^ ICD-O-3 morphology codes: 9383/1, 9391/3, 9392/3, 9393/3 and 9394/1; ^b^ ICD-O-3 morphology codes: 9470/3, 9471/3, 9472/3, 9474/3 and 9480/3; ^c^ ICD-O-3 morphology code: 9421/1; ^d^ ICD-O-3 morphology code: 9400/3; ^e^ Malignant: 5th digit of the morphology code of the IDC-O-3 equal 3; ^f^ Non-malignant: 5th digit of the morphology code of the IDC-O-3 < 3; ^g^ ICD-O-3 morphology codes: 9401/3, 9440/3, 9441/3, 9442/3, 9451/3 and 9508/3; ^h^ Malignant glioma: ICD-O-3 morphology code: 9380/3, optical tract excluded. ICD-O-3: International Classification of Diseases for Oncology, third ed. [26]. ^ Statistically significant trend.

**Table 7 cancers-15-05889-t007:** Astrocytomas (Subgroup IIIb of the ICCC-3) in children and adolescents, period 1991–2005. 5-year observed survival and follow-up by cohort of diagnosis and age group, and standardised survival for children. Log-rank results comparing the equality of survival distributions by age group, log-rank trend for observed survival, and annual percent change (APC) for trends in standardised survival of children.

Cohort of Diagnosis	Age Group					Log-Rank (*p*)
	Children			Adolescents		
	n	5-y Survival (%) (95% CI)		n	5-y Observed Survival (%) (95% CI)	
		Observed	Standardised			
1991–1995	145	80 (73; 86)	80 (73; 86)	40	73 (59; 86)	0.370
1996–2000	148	81 (75; 87)	82 (75; 88)	30	76 (61; 91)	0.520
2001–2005	123	89 (83; 94)	88 (83; 94)	28	64 (46; 82)	<0.001
Total	416					
Log-rank trend (*p*)		0.133			0.650	
Standardised survival trend APC (95% CI)			6 (−15; 31)			

ICCC-3: International Classification of Childhood Cancer, third ed. [25]; n: number of cases; 5 y: 5-year; (95% CI): confidence interval.

**Table 8 cancers-15-05889-t008:** Medulloblastomas ^a^ in children, period 1991–2005. 5-year observed survival by cohort of diagnosis and age group, and standardised survival in patients aged 0–14 years. Log-rank results comparing the equality of survival distributions by age group and log-rank trends for observed survival, annual percent change (APC) for standardised survival, and 5-year follow-up. Survival results are not shown for <10 cases, log-rank results are not shown when there are no survival results for all age groups, and log-rank for trend results are not shown by age group, nor are annual percent change, when there are no survival results for the three cohorts.

Age Group	Cohort of Diagnosis						Log-Rank Trend (*p*)
	1991–1995		1996–2000		2001–2005		
	n	5-y Observed Survival (%) (95% CI)	n	5-y Observed Survival (%) (95% CI)	n	5-y Observed Survival (%) (95% CI)	
0	2		2		2		
1–4	20	35 (14; 56)	19	42 (20; 64)	18	39 (16; 61)	0.967
5–9	24	50 (30; 70)	17	47 (23; 71)	20	60 (38; 82)	0.745
10–14	17	65 (42; 87)	10	80 (55; 100 *)	9		
0–14	63	48 (35; 60)	48	54 (40; 68)	49	55 (41; 69)	0.662
Standardised survival 0–14 years		46(34;58)		55 (41; 68)		57 (44; 69)	APC (95% CI)
							11 (−32; 80)

n: number of cases; 5 y: 5-year; (95% CI): confidence interval. ^a^ ICD-O-3 morphology codes: 9470/3, 9471/3, 9472/3, 9474/3 and 9480/3. ICD-O-3: International Classification of Diseases for Oncology, third ed. [26]. * Truncated upper limit.

**Table 9 cancers-15-05889-t009:** Malignant ^a^ and non-malignant ^b^ CNS tumours in children and adolescents, period 1991–2005. 5-year observed survival and follow-up by age group and behaviour. Log-rank results comparing the equality of survival distributions by behaviour and between children and adolescents.

Diagnostic Group	Children		Adolescents		Log-Rank (*p*)
	n	5-y Observed Survival (%) (95% CI)	n	5-y Observed Survival (%) (95% CI)	
Malignant ^a^	738	58 (55; 62)	140	54 (46; 63)	0.381
Non-malignant ^b^	234	87 (83; 91)	59	90 (82; 97)	0.562
Log-rank (*p*)		<0.001		<0.001	

CNS: central nervous system (Group III of the ICCC-3); ICCC-3: International Classification of Childhood Cancer, third ed. [25]; n: number of cases; 5 y: 5-year; (95% CI): confidence interval. ^a^ Malignant: 5th digit of the ICD-O-3 morphology code equals 3; ^b^ Non-malignant: 5th code of the ICD-O-3 < 3. ICD-O-3: International Classification of Diseases for Oncology, third ed. [26].

**Table 10 cancers-15-05889-t010:** CNS tumours in children (malignant and non-malignant). 5-year survival in Spain (2001–2005) and Europe (2000–2007) by ICCC-3 diagnostic subgroup and behaviour.

Diagnostic Subgroup (Group III of the ICCC-3)	5-y Survival (%) (95%CI)	
	Spain 2001–2005	Europe ^b^ 2000–2007
IIIa. Ependymomas and choroid plexus tumours ^a^	47 (30; 64)	70 (67; 72)
IIIb. Astrocytomas ^a^	89 (83; 94)	80 (79; 81)
IIIc. Embryonal ^a^	49 (37; 60)	57 (55; 59)
IIId. Other gliomas ^a^	30 (14; 46)	46 (43; 49)
IIIe. Other specified ^a^	85 (73; 97)	93 (91; 94)
IIIf. Unspecified ^a^	52 (32; 72)	64 (60; 67)
III. Total CNS ^c^	66 (60; 72) ^d^	71 (71; 72) ^e^
Behaviour		
Malignant ^c,f^	60 (53; 66)	57 (56; 58)
Non-malignant ^c,g^	88 (81; 95) ^d^	94 (94; 95) ^e^

CNS: central nervous system (Group III of the ICCC-3); ICCC-3: International Classification of Childhood Cancer, third ed. [25]; 5 y: 5-year; (95% CI): confidence interval. ^a^ Observed survival for both Spain and Europe; ^b^ European data from Gatta et al. 2017 [42]: pool of 26 countries; ^c^ Survival is age-standardised according to the European pool; ^d^ Four registries excluded from the estimates for total CNS and non-malignant tumours due to incomplete registration of non-malignant tumours; ^e^ Six countries excluded from the estimates for total CNS and non-malignant tumours due to incomplete registration of non-malignant tumours; ^f^ Malignant: 5th digit of the morphology code of the ICD-O-3 = 3; ^g^ Non-malignant: 5th digit of the morphology code of the ICD-O-3 < 3. ICD-O-3: International Classification of Diseases for Oncology, third ed. [26].

## Data Availability

The data supporting this study have been anonymised. Results are available, always in aggregated form, upon request and formal agreement, provided that there are technical and legal guarantees regarding the protection of personal data and the specific permission of the cancer registries concerned.

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
