# Peer review of "Childhood and Adolescent Central Nervous System Tumours in Spain: Incidence and Survival over 20 Years: A Historical Baseline for Current Assessment"

_cancers, 2023, doi:10.3390/cancers15245889_

Round 1
Reviewer 1 Report
Comments and Suggestions for Authors
The manuscript represents a very important and comprehensive dataset analyses. The statistical analyses is well conducted and the results are clearly presented. The following aspects should be addressed before further consideration:
Correc formatting errors in the affiliation and in the text.
Abstract: Define PBCRs, ICCC and other abbreviations.
Line 81: without improvement – over time? Please specify
Line 94: incidence rates instead of rates
Line 110-116: add references
Line 141: included cases
Line 158-160 and also 212-213: why just a subset pf the cases was used for survival studies?
Table 1: 0-14 years Navarre presents a relatively high percentage of Letha cases 14.9% compared to the other centers. 15-19 Balearic Islands also very high with 40% lethal cases. Inge general there is a high variability in lethality. Can you try and find an explanation why?
Do you have any treatment data?
Comments on the Quality of English Language
minor editing required by a native speaker
Reviewer 2 Report
Comments and Suggestions for Authors
This article is interesting, but some points need to be revised:
- As this paper collects a lot of data from different regions of Spain, it would be interesting to understand if there are differences in tumor incidence in different geographical areas. These important data do not emerge at the moment. Improve this part.
- I think it is better to merge same figures like 4a/4b/4c or 6a/6b or in a single figure, like figure 4 and figure 6
- Lines 911-912: "Survival in Spain is lower than the European 911 average, with no improvement" What do the authors attribute this lack of improvement to?
- Figures 4c and 5b are not present
Comments on the Quality of English LanguageMinor editing of English language required
Round 2
Reviewer 1 Report
Comments and Suggestions for Authors
The authors have reviewed the manuscript appropriately.
Comments on the Quality of English Languagejust minor typos